# Optimization of the Fermentative Production of *Rhizomucor miehei* Lipase in *Aspergillus oryzae* by Controlling Morphology

**DOI:** 10.3390/bioengineering9110610

**Published:** 2022-10-25

**Authors:** Chao Li, Dou Xu, Zhiyue Xiong, Yiming Yang, Guiwei Tian, Xuezhi Wu, Yonghong Wang, Yingping Zhuang, Ju Chu, Xiwei Tian

**Affiliations:** 1State Key Laboratory of Bioreactor Engineering, East China University of Science and Technology, Shanghai 200237, China; 2Wilmar Shanghai Biotechnology Research and Development Center Co., Ltd., Shanghai 200135, China

**Keywords:** *Aspergillus oryzae*, *Rhizomucor miehei* lipase, morphology, spore concentration, precision fermentation

## Abstract

Morphology plays an important role in the fermentation bioprocess of filamentous fungi. In this study, we investigated the controlling strategies of morphology that improved the efficiency of *Rhizomucor miehei* lipase (RML) production using a high-yield *Aspergillus oryzae.* First, the inoculated spore concentrations were optimized in seed culture, and the RML activity increased by 43.4% with the well-controlled mycelium pellets in both ideal sizes and concentrations. Then, the initial nitrogen source and agitation strategies were optimized to regulate the morphology of *Aspergillus oryzae* in a 5 L bioreactor, and the established stable fermentation system increased the RML activity to 232.0 U/mL, combined with an increase in total RML activity from 98,080 U to 487,179 U. Furthermore, the optimized fermentation strategy was verified by a high-yield *Aspergillus oryzae* and achieved an additional improvement of RML activity, up to 320.0 U/mL. Moreover, this optimized fermentation bioprocess was successfully scaled up to a 50 L bioreactor, and the RML activity reached 550.0 U/mL. This work has established a stable precision fermentation bioprocess for RML production by *A. oryzae* in bioreactors, and the controlling strategy developed in this study could potentially be extended to an industrial scale for RML production with high efficiency.

## 1. Introduction

As a typical interfacial enzyme, *Rhizomucor miehei* lipase (RML) can catalyze ester synthesis, ester exchange, ester hydrolysis, and other reactions in a non-aqueous environment [1,2]. Moreover, RML has been widely used in food processing, medicine, detergents, and other industries [3,4,5,6,7]. The wild-type *R. miehei* strain produces low levels of RML with unstable composition, which cannot satisfy the requirements for the industrial scale-up of RML production. However, the rapid development of synthetic biology tools and metabolic engineering strategies allows the construction of engineered heterologous expression systems including *Pichia pastoris* and *Aspergillus oryzae* for improved RML production. Although there have been many reports using *P. pastoris* as a host to produce RML with high enzymic activity [8], those protocols are difficult to be adopted widely in industrial environments due to food safety limitations. In contrast, *A. oryzae*, as a generally recognized as safe (GRAS) strain, can be successfully employed for the exogenous expression of RML. Unfortunately, the current enzymic activity of RML in *A. oryzae* is too low to guarantee successful commercialization [7]. Therefore, there is an urgent need to develop efficient methods to improve the production of RML by *A. oryzae*.

The liquid culture of filamentous fungi is often accompanied by variations in morphology. These changes can significantly affect the rheology, mass transfer, and mixing characteristics of the broth, and thus, exerts a huge impact on the synthesis of the final product [9,10,11]. In general, the morphology of filamentous fungi could be divided into three types: dispersed mycelia, clumps, and dense pellets [12]. Moreover, the final products can exhibit significant differences in the optimal morphology [13,14]. For example, pellets are considered to be the optimal form in the production of glucoamylase by *A. niger*. In contrast, mycelia are known to be the most suitable form for citric acid fermentation by *A. niger* [15].

The production of RML by *A. oryzae* is a complex fermentation bioprocess that is often associated with several issues, such as sticking to the wall of the bioreactor and aggregation on the top of the broth due to the overgrowth of mycelia. These factors create a heterogeneous fermentation environment that affects the mass transfer and mixing characteristics of the broth, and is prone to liquid escape, thus, causing an early end to fermentation. Moreover, the effective volume of the broth is greatly reduced, thereby reducing production efficiency. The morphology of filamentous fungi can usually be regulated by physical and physiological approaches during the fermentation process. The former option includes the adjustment of agitation [11], the design of the bioreactor structure [16], and the addition of microparticles [17,18], while the latter option can be accomplished by the optimization of spore concentration [19], the type, and the concentration of the nitrogen source [20,21].

In this study, we devised different approaches to regulate morphology to improve RML production by *A. oryzae*. Firstly, the effects of different inoculated spore concentrations were examined on morphology and RML production in shake flasks. Secondly, nitrogen source supply and agitation control strategies were optimized in a 5 L bioreactor to achieve suitable morphologies for RML production. Finally, the process and morphology control strategy was verified by a high-yield *A. oryzae* strain in a 5 L bioreactor and subsequently scaled up to a 50 L bioreactor.

## 2. Materials and Methods

### 2.1. Microorganisms, Media, and Culture Conditions

Original *A. oryzae* CGMCC18825 was kindly provided by Wilmar Biotechnology R&D Center Co., Ltd., Shanghai, China. High-yield strains of *A. oryzae* CGMCC3.16200 (III3-5C-1) were obtained by atmospheric and room temperature plasma (ARTP) mutagenesis [22]. Seed medium (in g/L) consisted of corn dextrin (Sinopharm) 20; dry corn pulp powder (Sinopharm) 10; and yeast extract (Angelyeast) 10, as well as polyether defoamer (BASF) 1; KH_2_PO_4_ (Sinopharm) 1.244; MgSO_4_·7H_2_O (Sinopharm) 0.249; and Na_2_HPO_4_·12H_2_O (Sinopharm) 6.0.

*A. oryzae* spores were cultured in potato glucose agar (PDA) medium at 28 °C for 2 days. Then, a single fast-growing colony growing was selected and cultured for 7 days. Spores (cyan-green) were washed off using sterile water and counted with a blood count board. Then, the spores were diluted with sterile water to an appropriate concentration and stored at 4 °C. Four spore suspensions with the concentration of 10^4^, 10^5^, 10^6^, and 10^7^ (in spores/mL) were prepared by gradient dilution for preliminary optimization in seed culture. Furthermore, other four-spore suspensions with concentrations of 7 × 10^4^, 9 × 10^4^, 3 × 10^5^, and 5 × 10^5^ (in spores/mL) were prepared for further optimization. In seed culture, 300 μL of spore suspension was transferred into the shake flask and cultured at 28 °C with a rotation speed of 150 rpm for 28 h. For the fermentation in a shake flask, the seed liquid was inoculated into a 250 mL shake flask with a working volume of 50 mL (10% inoculum) and cultured at 28 °C with a rotation speed of 200 rpm.

A 5 L bioreactor (Shanghai Guoqiang Bioengineering Equipment Co., Ltd., Shanghai, China) equipped with two standard Rushton impellers (diameter of 60 mm) was used to establish a basic bioprocess in bioreactor level. The initial fermentation medium contained (in g/L) corn dextrin (Sinopharm) 50; soybean peptone (Sinopharm) 30; yeast extract (Angelyeast) 4; polyether defoamer (BASF) 1; KH_2_PO_4_ (Sinopharm) 1.12; MgSO_4_·7H_2_O (Sinopharm) 0.227; and Na_2_HPO_4_·12H_2_O (Sinopharm) 5.33. All media (except for corn dextrin solution, which was sterilized at 115 °C for 20 min) were sterilized at 121 °C for 30 min. The seeds were prepared by the above method. The seed liquid was inoculated into a 5 L bioreactor (initial working volume of 2.5 L) with a 10% inoculum at 28 °C for 144 h. The aeration rate was 1 vvm and the initial agitation speed was 300 rpm. Samples were taken every 24 h to test the off-line parameters including RML activity, protein content, dry cell weight, and mycelial morphology. The medium, sterilization, and cultivation conditions, and the process control in a 50 L bioreactor (Shanghai Guoqiang Bioengineering Equipment Co., Ltd., Shanghai, China) were similar to those in the 5 L bioreactor.

### 2.2. Optimization of Nitrogen Source Concentration and Agitation

To avoid hyphae growth and adherence in the early stage of fermentation, we optimized the ratio of nitrogen source in the base medium to that used for feeding. The initial concentrations of the nitrogen source were set as 10% (3 g/L), 20% (6 g/L) and 50% (15 g/L) of the original amount and the remaining were fed intermittently. During fermentation, the oxygen uptake rate (OUR) was determined by a process mass spectrometer (MAX300-LG, Extrel, USA), as described by Chen et al. [23]. The nitrogen sources were fed at a volume of 150 mL each time when the OUR was reduced to 5.0 mmol/L/h.

In terms of the optimization of agitation strategy, an initial process was defined as Mode 1, while the optimized process was defined as Mode 2. To control the morphology and provide enough oxygen supply, the agitation speed in Mode 2 was significantly lower than Mode 1 in the early stage of fermentation, while the value in Mode 2 was significantly higher than Mode 1 in the late stage (Figure A1).

### 2.3. Detection of RML Activity, Protein, Sugar, and DCW

RML activity was assayed by the 4-Nitrophenyl palmitate (*p*-NPP) method [24]. In brief, *p*-NPP solution and substrate buffer solution (containing (g/L) NaH_2_PO4·2H_2_O 0.413, Na_2_HPO_4_ 16.975, sodium deoxycholate 2.3, and acacia 1.1) were mixed in a ratio of 1: 9 (*v*/*v*). Then, 2.4 mL of the mixture was added to the test tube and preheated for 3 min at 37 °C. Subsequently, 100 μL of diluted fermentation supernatant was added to the sample tube and 100 μL of boiled and denatured (100 °C for 5 min) sample solution was added to the control tube as a blank. All samples were tested with three parallels. The reaction was terminated by placing in a water bath at 37 °C for 15 min followed by the addition of 2 mL of 95% ethanol. Then, the solutions were centrifuged at 4000 r/min for 5 min. The absorbances of the sample and the blank were tested at 410 nm.

The protein content in the broth was assayed by the Coomassie brilliant blue method [25]. First, 100 μL of diluted fermentation supernatant was added to the sample tube with distilled water as a blank. Then, 5 mL of Coomassie brilliant blue dye reagent was added to the tube, and then evenly mixed and placed at room temperature for 5 min. The absorbance of the mixture was measured at 595 nm.

The concentration of residual sugar in the broth was measured using the following steps [26]. First, 1 mL of diluted fermentation supernatant was added into a 10 mL colorimetric tube and hydrolyzed with 1.5 mL of 3 M HCl in a boiling water bath for 20 min. Second, 1.5 mL of 3 M NaOH solution and 1.5 mL of dinitro salicylic acid (DNS) reagent were added to the tube, well mixed, and kept in a boiling water bath for 5 min. Finally, the absorbance of the sample was measured at 550 nm.

Dry cell weight (DCW) was measured. The cells were collected by suction filtration and washed with two volumes of distilled water, and then dried at 80 °C to a constant weight.

### 2.4. Determination of Cell Morphology

The agarose fixation method [27] with slight modifications was used to determine the cell morphology. Firstly, 1 mL of cell suspension was fixed with 40% (*v*/*v*) formaldehyde and 60% (*v*/*v*) ethanol. Then, the suspension was diluted with sterile water (1:9) and poured into the plate, and photographs of uniform size were taken under a daylight lamp based on the diameter of the plates. Image-Pro Plus software (Media Cybernetics, Rockville, MD, USA) was used to process the images and calculate the mean diameter and the concentration of the pellets (the number of pellets contained in 1 mL of fermentation broth).

### 2.5. Data Analysis

Total RML activity (U) was calculated using Equation (1).
Total RML activity (U) = RML activity (U/mL) × working volume (L) × 1000(1)

The working volume represents the volume of flowable liquid in a 5 L bioreactor.

All the experiments were performed in triplicates and mean values with standard deviations were reported using Microsoft Excel. A one-way ANOVA was performed followed by Duncan’s post hoc analysis at *p* = 0.05 in SPSS Software.

## 3. Results

### 3.1. Effects of Inoculated Spore Concentration on Cell Morphology and RML Production in Shake Flasks

The effects of different inoculated spore concentrations (10^4^, 10^5^, 10^6^, and 10^7^ spores/mL) on cell growth and morphology were investigated in seed culture. After culturing for 28 h, it was found that the DCW increased with the rising inoculated spore concentrations, while the residual sugar concentration decreased gradually (Table 1). Notably, *A. oryzae* exhibited completely different morphologies: large pellets (S1, 10^4^ spores/mL); radial pellets (S2, 10^5^ spores/mL); aggregated mycelia (S3, 10^6^ spores/mL); and relatively dispersed mycelia (S4, 10^7^ spores/mL) (Figure A2). It seems that it is easier to form pellets at a low inoculated spore concentration, while it is easier to form mycelia at a high inoculated spore concentration, which is consistent with the findings reported by Teng et al. [10].

Then, the seeds were inoculated into a fermentation medium and cultured for 5 days in shake flasks. The DCW and sugar consumption in 1 × 10^7^ spores/mL were higher than others, but the highest RML activity was achieved in the experiment with 10^5^ spores/mL (209.0 U/mL) (Table 2). Interestingly, all four conditions resulted in the formation of pellets after 5 days of culture. However, the concentration and size of the pellets were significantly different (Figure A2). With the increase in inoculated spore concentration, the pellet concentration gradually increased from 22 pellets/mL to 2154 pellets/mL, while the mean diameter of pellets decreased from 2.99 mm to 0.54 mm. With the inoculated spore concentration of 10^5^ spores/mL, the RML activity was relatively higher than others, corresponding to a pellet concentration of 1495 pellets/mL and a mean pellet diameter of 0.79 mm, respectively (Table 2). The results indicated that there might be an optimal concentration and size of pellets for RML production by *A. oryzae*.

To further optimize the inoculated spore concentration in seed culture, the effects of spore concentrations of 7 × 10^4^, 9 × 10^4^, 3 × 10^5^, and 5 × 10^5^ spores/mL on RML fermentation were investigated subsequently. It is observed that the highest RML activity (217.5 U/mL) was reached at a spore concentration of 3 × 10^5^ spores/mL (Table 3). Moreover, the morphological analysis showed that the concentration and mean diameter of the pellets were 1516 pellets/mL and 0.77 mm, respectively. Although RML activity improved from 151.7 U/mL (1 × 10^7^ spores/mL) to 217.5 U/mL (3 × 10^5^ spores/mL) in seed culture, the protein content was enhanced from 0.79 g/L to 1.06 g/L correspondingly, demonstrating that there was no significant difference in the specific RML activity. As a result, the specific RML activity remained at a relatively stable level of 0.21 U/g protein (Figure 1). The results indicated that the inoculated spore concentration could change the morphology of *A. oryzae* and subsequently affect the metabolic environment and increase the production of RML.

### 3.2. Optimization of the RML Production Based on Morphology Control in 5 L Bioreactor

#### 3.2.1. Morphology Control by Optimization of Nitrogen Source Supply Strategy

When the fermentation was carried out in a 5 L bioreactor with the original medium, the mycelia stuck to the wall of the bioreactor and aggregated on the top of the bioreactor, resulting in a significant reduction in the effective working volume. Liquid escape occurred after 48 h, thereby leading to the early termination of fermentation. The total RML activity was only 98,080 U. This emphasized the need to establish a stable fermentation system to improve the total RML activity.

To control cell growth in the early stage of fermentation and establish a stable fermentation system, the original nitrogen source concentrations were reduced to 10%, 20%, and 50% of the original amount, and the rest were fed during fermentation. Many solid mycelia aggregates appeared after 72 and 96 h of culture under the conditions of 20% and 50% initial nitrogen source (Figure 2), resulting in a reduction in the effective working volume and liquid escape. Although the reduction in the initial nitrogen source concentration improved the rheological behavior of the broth to a certain extent, the fermentation was unable to maintain a normal duration (144 h). In contrast, pellets were formed in the early stage of fermentation under the 10% initial nitrogen source (Figure 2), and the process maintained a normal duration. The results indicated that the issues of mycelium adherence above were solved, and a stable fermentation process was established. By comparing cell growth, RML production, and protein synthesis under the three conditions, it is found that a reduction in the initial nitrogen source retarded the cell growth in the early stage of fermentation (before 24 h) (Figure 3A). Furthermore, the production of RML and protein synthesis in the early stage also decreased with the reduction in initial nitrogen source (Figure 3B,C), but the product synthesis was prolonged. Moreover, with a 10% initial nitrogen source, the effective working volume reached 2.0 L at the end of fermentation, which was approximately twice as much as that with 20% and 50% initial nitrogen sources. The OUR was relatively lower at a high initial nitrogen source (Figure 3D), which could be attributed to the increase in mass transfer resistance due to the mycelia aggregation and poor rheological behavior. With the 10% initial nitrogen source, the RML activity reached 188.7 U/m, and the total RML activity was 377,400 U, which was 4.9- and 3.1-fold of that in the 20% and 50% initial nitrogen sources.

#### 3.2.2. Morphology Control by Optimization of Agitation Strategy

Different agitation speeds in the bioreactor could generate a variant shear environment and oxygen supply, which would significantly affect the hyphae growth and morphology in *A. oryzae* fermentation. Morphological analysis under the 10% initial nitrogen source condition showed that the mean diameter of the pellets increased first and then decreased, but the pellet concentrations showed an opposite trend (Table 4). During the early stage of fermentation, the rapid cell growth and elongation of the mycelia led to a gradual enlargement of pellets. However, in the middle and late stages, the increase in agitation accompanied by higher shear force could result in the breakup of the large pellets and the detachment of hyphae from the pellets. With the increase in shear force, the large pellets formed in the early stage were gradually shredded, and the hyphae on the surface of the pellets were removed, which jointly promoted the decrease in pellet diameter and the increase in pellet number [28]. Unfortunately, compared to the optimal morphology observed in the shake flask, the mean diameter of pellets (1.01 mm) in the 5 L bioreactor was larger, and the pellet concentration was significantly lower (994 pellets/mL). Furthermore, it was found that the pellet diameter experienced a rapid increase in the middle stage of the fermentation (48–72 h). Therefore, to control the morphology and achieve higher RML activity, a lower agitation speed was adopted during the middle phase to decrease the oxygen supply and inhibit the pellet growth, while a higher agitation speed was adopted during the late phase to increase the shear force and obtain a larger number of smaller pellets.

Compared to the agitation strategy of Mode 1, the cell growth in Mode 2 was inhibited to a certain extent during the middle phase of fermentation (Figure 4A), and the OUR level was relatively lower (Figure 4B). Consequently, RML production and protein synthesis were also reduced (Figure 4C,D). However, during the late phase of fermentation, the agitation speed in Mode 2 was higher, which resulted in higher shear force and a better oxygen supply (a higher OUR). The RML production and protein synthesis in Mode 2 were significantly higher than those in Mode 1. The RML activity in Model 2 reached 232.0 U/ mL at the end of fermentation (144 h), which was enhanced by 22.9% compared to Mode 1. Moreover, the effective working volume in Mode 2 also increased, leading to an increase in total RML activity by 29.1% (Table 4). Furthermore, in Mode 2, the size of the pellets was significantly reduced, and the concentration of pellets increased as expected during the middle and late phases of fermentation. The mean diameter of the pellets decreased from 1.01 mm to 0.97 mm (Figure 4E), while the pellet concentration increased from 994 to 1072 pellets/mL at 144 h (Figure 4F).

### 3.3. Strategy Verification Using a High-Yield Strain in a 5 L Bioreactor

Four high-yield strains were screened in our previous study by ARTP mutagenesis: I7-6D-7, V7-6C-3, II4-4B-5, and III3-5C-1 [22], and the problems met in the original strain also appeared in the high-yield strains. Preliminary experiments in shake flasks under optimized conditions showed that III3-5C-1 exhibited the highest RML activity (307.4 U/mL), which was 50.4% higher than the original strain (Figure 5A). Moreover, there was no significant change in the RML activity after three consecutive subcultures (Figure 5A). Thus, strain III3-5C-1 was selected for the verification of the above strategies in 5 L bioreactors.

The cell growth and morphology of III3-5C-1 presented a similar trend to the original strain during the whole fermentation. However, the productivity of RML during the late phase of fermentation was significantly higher, leading to a final RML activity of 320.0 U/mL (Figure 5B), which was 37.9% higher than that of the original strain. Otherwise, the protein concentration had no significant difference (Figure 5C), indicating that the specific RML activity of III3-5C-1 was significantly higher than the original strain, especially in the late phase of fermentation (Figure 5D).

### 3.4. Strategy Verification and Scale-Up in a 50 L Bioreactor

According to the above results, the morphology control strategy was verified and scaled up using the high-yield strain in a 50 L bioreactor. After 144 h of fermentation, the RML activity in the 50 L bioreactor was increased to 550.4 U/mL (Figure 6A), and protein concentration reached 2.5 g/L (Figure 6B), which were 72% and 56% higher than those in the 5 L bioreactor (Figure 5), respectively. Meanwhile, the concentration of pellets decreased first and then increased, while the average diameter of pellets increased first and then decreased, indicating that the variation of morphology in the 50 L bioreactor was consistent with that in the 5 L bioreactor (Figure 6C,D). However, In the 50 L bioreactor, the concentration of pellets was significantly higher than that in the 5 L bioreactor, and the size of pellets was significantly lower than that in the 5 L bioreactor. Interestingly, at the end of fermentation in the 50 L bioreactor, the concentration of pellets reached 1554 pellets/mL and the average diameter of pellets was reduced to 0.76 mm, which was very close to the optimal concentration of pellets (about 1516 pellets/mL) and the average diameter of pellets (about 0.77 mm) observed in the shake flask. As a similar agitation strategy was adopted in the 50 L bioreactor (Figure 6E); a better oxygen supply (Figure 6F) and higher shear force conditions could be formed in the 50 L bioreactor, leading to better control of the morphology and improvement of the RML activity.

## 4. Discussion

Most of the previous reports related to RML production were carried out in a heterologous expression system by genetic engineering. *Pichia pastoris* has been commonly used to produce RML, and the maximum RML activity reached 116 U/mL with the *p*-NPP (*p*-nitrophenol palmitate) method [6,8,29]. Although *A. oryzae* is known as GRAS, the reported RML activity was only 2.5 U/mL [7]. Moreover, there had been few reports on RML production in a bioreactor by *A. oryzae*, of which one of the possible causes was the difficulties in controlling the morphology of filamentous fungi in fermentation. In this work, by deploying a systematic strategy for morphology control involving inoculated spore concentration, nitrogen source supply and agitation optimization, an RML activity of 550.0 U/mL by *A. oryzae* in a 50 L bioreactor was reported for the first time. Based on the above work, a stable and uniform fermentation system was established, which effectively solved the problems involving adherent growth, aggregation, and the escape of liquid.

Controlling morphology by the application of physical and physiological approaches to further increase production efficiency has always been a significant and challenging aspect of filamentous fungal fermentation. Previous studies have shown that the inoculated spore concentration, the types and concentrations of carbon and nitrogen sources, and agitation, could significantly affect the morphology [30,31,32,33]. Wang et al. [19] found that when the concentration of spores was less than 10^6^/mL, *Trichoderma harzianum* grew in the form of pellets. The lower the concentration of spores, the larger the pellets and the smaller the biomass. When the concentration of spores exceeded 10^6^/mL, mycelia were the predominant morphology and the biomass decreased. It is worth noting that pellets with a large diameter would slow down the mass and oxygen transfer, further affecting nutrient utilization and cellular metabolism, and finally resulting in a low yield. However, if the pellets are too small or dispersed, the viscosity of the broth would increase, which would hinder oxygen transfer and lead to a reduction in yield [34].

In this study, the appropriate size and number of pellets for efficient RML production by *A. oryzae* were investigated by optimizing the concentration of spores. This concept was then used to regulate morphology in the bioreactors. Nitrogen sources could not only control cell growth, but also played an important role in lipase production [21,35]. By reducing the initial nitrogen source, the morphology was successfully controlled during fermentation, and the issues including low working volume and liquid escape induced by adherent growth were solved. Furthermore, with the optimization of agitation, the diameter and concentration of the pellets were optimized, and the RML activity was further improved. The oxygen supply and shear force caused by different agitation strategies could affect the morphology significantly. Lower agitation speed in the middle phase resulted in an inferior oxygen supply, which could inhibit pellet growth. However, higher agitation speed in the late phase resulted in higher shear force and better oxygen supply, which were more conducive to the formation of small-diameter pellets and lipase production [27]. Finally, a high-yield strain Ⅲ3-5C-1 was selected to verify the strategy in 5 L and 50 L bioreactors. An RML activity of 550.0 U/mL by *A. oryzae* was achieved in a 50 L bioreactor, which was the highest level reported in the literature. The control strategies would provide a solid foundation for the industrial production of RML. However, more detailed investigations about the relationships between inoculation load and morphology, as well as the quantitative relationships between the morphology and shear force, would be helpful for the rational regulation of the morphology and further improvement of RML productivity.

## 5. Conclusions

In this work, a systematic strategy has been deployed for morphology control in the precision fermentation bioprocess of *A. oryzae* for RML production. The controlling factors include the inoculated spore concentration, nitrogen source supply, and agitation optimization, and a stable bioprocess for RML fermentation by *A. oryzae* in a 50 L bioreactor has been reported for the first time. Firstly, the RML activity increased from 151.7 U/mL to 218.6 U/mL in the shake flask by controlling the inoculated spore concentration at 3 × 10^5^ spores/mL. In a 5 L bioreactor, a stable fermentation system was established with the controlled morphology by the reduction in the initial nitrogen source and the optimization of agitation strategy, leading to an increase in the total RML activity from 98,080 U to 487,179 U. Furthermore, a high-yield *A. oryzae* strain was selected to verify this strategy, leading to a further improvement of RML activity (320.0 U/mL) and total RML activity (639,910 U). Finally, the optimized process was scaled up to a 50 L bioreactor, and the RML activity reached 550.0 U/mL. This work established a stable precision fermentation process for RML production by *A. oryzae* in bioreactors. The proposed process control strategy in this study is promising and expected to scale up to an industrial scale for the highly efficient production of RML.

## Figures and Tables

**Figure 1 bioengineering-09-00610-f001:**
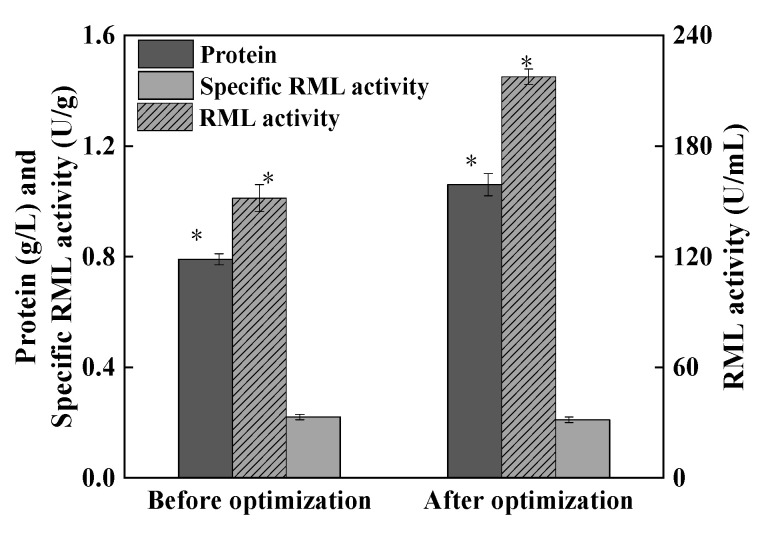
Comparison of RML production performance before and after the optimization of inoculated spore concentration in shake flasks. * indicates *p* < 0.05, data expressed as the mean ± standard deviation (n = 3 per group, per time point).

**Figure 2 bioengineering-09-00610-f002:**
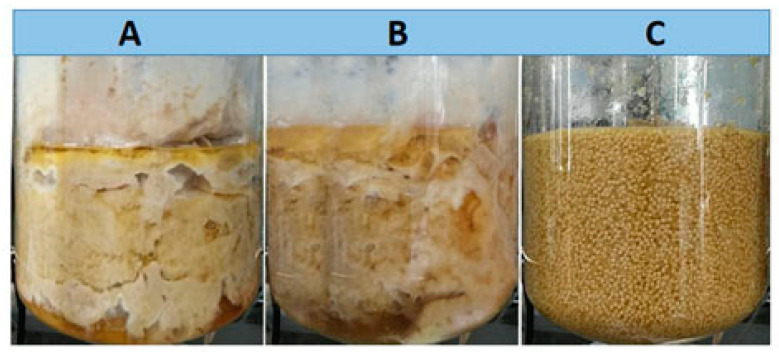
Mycelium morphologies under 50% (**A**), 20% (**B**), and 10% (**C**) of initial nitrogen source concentration in 5 L bioreactor.

**Figure 3 bioengineering-09-00610-f003:**
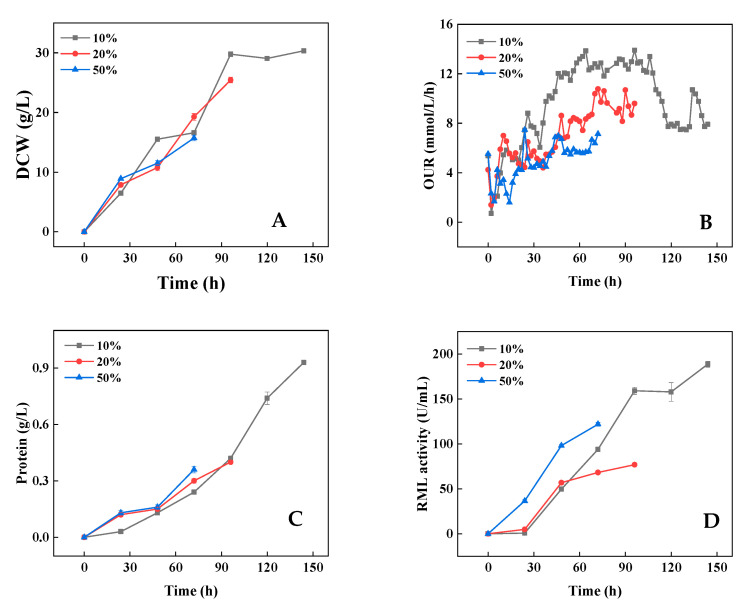
Cell growth (**A**), OUR (**B**), protein synthesis (**C**), and RML production (**D**) under different initial nitrogen source conditions. Data expressed as the mean ± standard deviation (n = 3 per group, per time point).

**Figure 4 bioengineering-09-00610-f004:**
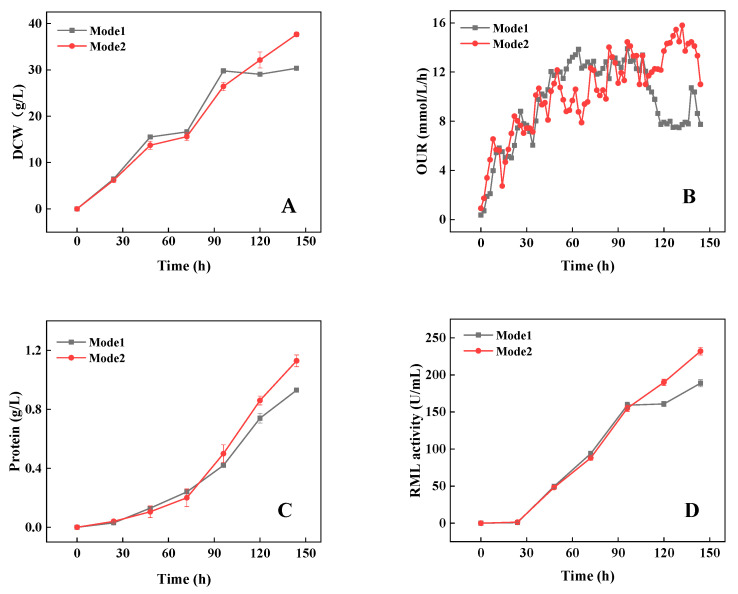
Comparison of DCW (**A**), OUR (**B**), protein (**C**), RML activity (**D**), average pellet diameter (**E**), and pellet concentration (**F**) with different agitation strategies. * indicates 0.01 < *p* < 0.05, ** indicates *p* < 0.01. Data expressed as the mean ± standard deviation (n = 3 per group, per time point).

**Figure 5 bioengineering-09-00610-f005:**
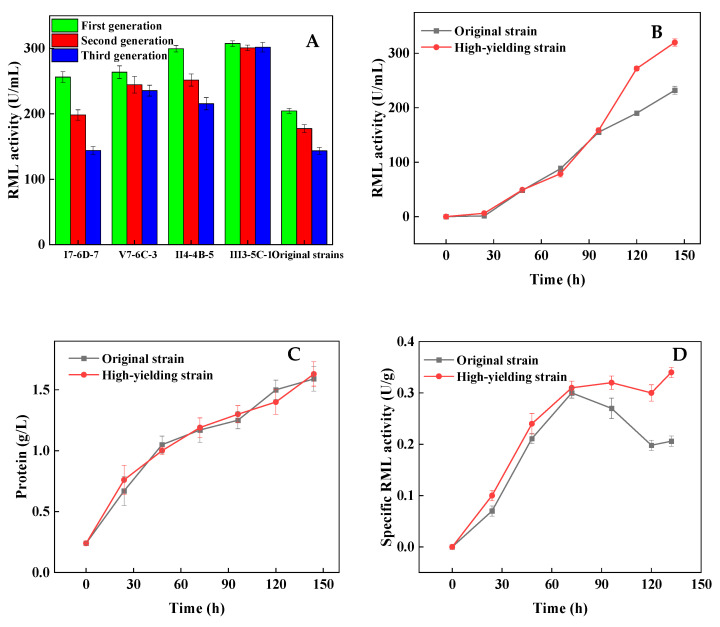
Comparison of the RML activity by different strains (**A**) in shake flasks, and the comparison of RML activity (**B**), protein concentration (**C**), and specific RML activity (**D**) between the high-yield strain and original strain in 5 L bioreactor. Data expressed as the mean ± standard deviation (n = 3 per group, per time point).

**Figure 6 bioengineering-09-00610-f006:**
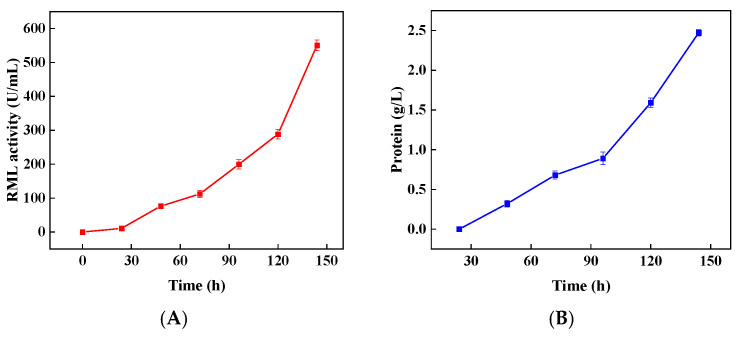
The RML activity (**A**), protein concentration (**B**), pellet concentration (**C**), pellet diameter (**D**), agitation strategy (**E**), and OUR (**F**) in 50 L bioreactor. * indicates 0.01 < *p* < 0.05, ** indicates *p* < 0.01. Data expressed as the mean ± standard deviation (n = 3 per group, per time point).

**Table 1 bioengineering-09-00610-t001:** Effect of different inoculated spore concentrations on cell growth and morphology in seed culture for 28 h. Data expressed as the mean ± standard deviation (n = 3 per group, per time point).

Spore Concentration (spores/Ml)	Residual Sugar (g/L)	DCW (g/L)	Morphology
1 × 10^4^	19.9 ± 0.3	7.8 ± 0.3	Larger pellets (S1)
1 × 10^5^	18.0 ± 0.5	8.8 ± 0.2	Radiate mycelium pellets (S2)
1 × 10^6^	12.6 ± 0.4	13.3 ± 0.2	Clustered mycelium (S3)
1 × 10^7^	10.4 ± 0.2	14.2 ± 0.2	Dispersed mycelia (S4)

**Table 2 bioengineering-09-00610-t002:** Effect of different morphologies on cell growth and metabolism in fermentation culture for 5 days in shake flasks. Data expressed as the mean ± standard deviation (n = 3 per group, per time point).

Spore Concentration (spores/mL)	RML Activity (U/mL)	Residual Sugar (g/L)	DCW (g/L)	Concentration of Mycelium Pellets (pellets/mL)	Average Diameter (mm)
1 × 10^4^	106.9 ± 1.3	1.9 ± 0.1	18.1 ± 0.4	22 ± 2.2	2.99 ± 0.17
1 × 10^5^	209.0 ± 0.7	1.7 ± 0.1	26.7 ± 0.4	1495 ± 5.0	0.79 ± 0.02
1 × 10^6^	181.1 ± 0.9	1.5 ± 0.1	27.7 ± 0.3	1874 ± 14.7	0.64 ± 0.02
1 × 10^7^	151.7 ± 0.9	1.4 ± 0.1	28.3 ± 0.1	2145 ± 77.2	0.54 ± 0.13

**Table 3 bioengineering-09-00610-t003:** Detailed optimization of inoculated spore concentration in shake flasks. Data expressed as the mean ± standard deviation (n = 3 per group, per time point).

Spore Concentration (spores/mL)	Average Diameter (mm)	Concentration of Mycelium Pellets (pellets/mL)	RML Activity (U/mL)
7 × 10^4^	1.05 ± 0.1	1030 ± 9.1	164.6 ± 1.0
9 × 10^4^	0.81 ± 0.1	1232 ± 43.0	166.4 ± 0.7
3 × 10^5^	0.77 ± 0.1	1516 ± 45.5	217.5 ± 1.1
5 × 10^5^	0.72 ± 0.1	1605 ± 10.1	204.4 ± 1.2

**Table 4 bioengineering-09-00610-t004:** Pellet concentrations, pellet average diameters, and total RML activities with different agitation strategies. Data expressed as the mean ± standard deviation (n = 3 per group, per time point).

	Mode 1		Mode 2	
Time (h)	Pellet Concentrations (pellets/mL)	Average Diameter (mm)	Pellet Concentrations (pellets/mL)	Average Diameter (mm)
12	270 ± 4	0.97 ± 0.01	262 ± 2	0.98 ± 0.02
24	60 ± 2	1.80 ± 0.12	71 ± 3	1.77 ± 0.12
48	122 ± 6	2.35 ± 0.11	174 ± 5	1.92 ± 0.11
72	206 ± 2	1.98 ± 0.10	264 ± 3	1.66 ± 0.13
144	994 ± 4	1.01 ± 0.01	1072 ± 2	0.97 ± 0.02
Total RML activity (U)	377,466 ± 1793		487,179 ± 2812	

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
