# Peer review of "Optimization of the Fermentative Production of Rhizomucor miehei Lipase in Aspergillus oryzae by Controlling Morphology"

_bioengineering, 2022, doi:10.3390/bioengineering9110610_

Round 1

Reviewer 1 Report

The paper by Li et al is interesting with regards to control of morphology of recombinant RMS-producing A. oryzae. Nontheless some major issues has to be addressed in order to recomment publication.

The tile should be more descriptive of the work, that is mainly on the control of morphology and the improvement of rheological properties of A. oryzae cultures.

Also the aim of the work, in the introduction should be better explained, in particular with regards to the utilization of mutants (see below).

The statistical analysis as weel as the presentation of statistically significant differences must be improved.

The text should be revised for clarity and style: an impersonal style at past tense should be utilized throughout the whole manuscript (e.g. lines 131136). The abstract and the text should be checked for typos.

More in detail:

Materials and methods

Line 75 and 90: yeast powder is yeast extract? What defoamer was utilized? Please provide the manufacturer.

Line 77: coated?

Line 79: eluted? Should be collected?

Line 81: Four spore suspensions

Line 99-104: Please rephrase for clarity and to enable experiment reproducibility. please replace nitrogen source with soybean peptone. It is not clear what 687.5, 625.0 and 437.5 g/L concentrations are. If they are the peptone concentration in 150 mL pulses, it should be better explained. What was the volume of the culture to which pulses were fed? The utilization of OUR to manage the feeding should be more breiefly and clearly described.

107: when talking about shear rate, please provide provide the number, the type and the dimension of the impellers with which the bioreactor was equipped.

Results

Section 3.1

The authors report different mycelia morphology in response to different inoculation load. They also report different sugar and DCW concentrations, indicating that 28 h cultures were in very different advancement status along the growth curve (the higher the inoculation load, the higher the biomass generated and the substrate consumed). Is it possible that morphology depends on incubation time, but the authors interpreted their observation as an effect of inoculation load? Is there any information on morphology changes during incubation of cultures?

Referring to 10^7 spores/mL as the control is questionable, as it is just a concentration as the others. Statistical comparison among the 4 initial spore concentrations should be calculated (e.g. with ANOVA with tukey post hoc) presented and presented in tables 1, 2, and 3 with superscripts that indicate significantly significant differences.

Tables 1, 2, 3: the legend should indicate the hours of the cultures. In table 2 it is not clear why the values are reported as function of S1, S2, S3, and S4 and not as function of 10^4, 10^5, 10^6, and 10^7 spores/mL.

187. What was the rational of selecting these values of spores/ml? It is not clear in what the optimization procedure consisted and the level of statistical significance of differences.

Figure 1: presumably values are means +/- SD. Please clarify and add indication of statistical significance

Section 3.2

Figure 2 Legend: it is better to replace “rheological properties of fermentation broth” (that would implicate some quantitative result) with “mycelium morphology in cultures”.

Section 3.2.2 could start at line 242

Table 4. It seems that the columns Spores concentrations should be entitled Pellets concentrations. The table and data description lacks statistical analysis, at least for comparing Mode 1 and Mode 2.

Figures 3 and 4. Is there any information on the trend of the carbon source? It would be useful to better comprehend the status of the culture during the course of the fermentation run.

Section 3.3

It should be better explained in the introduction the reason of reporting this section, that has not a lot to do the rest of the study that was carried out with a not-improved A. oryzae. The authors should clearly indicate was already known on the improved strains and why they included them in this manuscript that is mostly focused on the rheological behavior and in control of morphology. This section could be relevant if the mutants exhibited some morphology or behavior (e.g. formation of pellets) that could justify their increased performance.

Author Response

Point 1: The tile should be more descriptive of the work, that is mainly on the control of morphology and the improvement of rheological properties of A. oryzae cultures.

Response 1: Thanks for your suggestion, it’s a great help to improve the manuscript. we have changed the tittle and focused on morphology control.

Point 2: Also the aim of the work, in the introduction should be better explained, in particular with regards to the utilization of mutants (see below).

Response 2: We have edited the introduction based on helpful comments from the reviewer.

Point 3: The statistical analysis as well as the presentation of statistically significant differences must be improved.

Response 3: Agreed. We have improve the statistical analysis, especially in Figure 4 and Figure 6.

Point 4: The text should be revised for clarity and style: an impersonal style at past tense should be utilized throughout the whole manuscript (e.g. lines 131136). The abstract and the text should be checked for typos.

Response 4: Agreed. We have made a comprehensive proofreading for the manuscript and asked a native speaker to modify the text.

Point 5: Line 75 and 90: yeast powder is yeast extract? What defoamer was utilized? Please provide the manufacturer.

Response 5: Yes, we used yeast extract (Angelyeast) and polyether defoamer (BASF). The information has been added into the revised version.

Point 6: Line 77: coated?

Response 6: Sorry, it should be “cultured”. The description has been changed in the revised version.

Point 7: Line 79: eluted? Should be collected?

Response 7: Sorry, what we mean is washing off the spores with sterile water. The description has been changed in the revised version.

Point 8: Line 81: Four spore suspensions

Response 8: Yes, it should be “suspensions”

Point 9: Line 99-104: Please rephrase for clarity and to enable experiment reproducibility. please replace nitrogen source with soybean peptone. It is not clear what 687.5, 625.0 and 437.5 g/L concentrations are. If they are the peptone concentration in 150 mL pulses, it should be better explained. What was the volume of the culture to which pulses were fed? The utilization of OUR to manage the feeding should be more briefly and clearly described.

Response 9: Sorry to confuse you, we have changed description about nitrogen source supply and make it more clear about the feding strategy in part 2.2.

Point 10: 107: when talking about shear rate, please provide the number, the type and the dimension of the impellers with which the bioreactor was equipped.

Response 10: Agreed. We used two standard Rushton impellers with a diameter of 60 mm. The information has been added to part 2.1.

Point 11: The authors report different mycelia morphology in response to different inoculation load. They also report different sugar and DCW concentrations, indicating that 28 h cultures were in very different advancement status along the growth curve (the higher the inoculation load, the higher the biomass generated and the substrate consumed). Is it possible that morphology depends on incubation time, but the authors interpreted their observation as an effect of inoculation load? Is there any information on morphology changes during incubation of cultures?

Response 11: We have observed the changes of morphology during the whole fermentation, and found that the morphologies were totally different under different inoculation load conditions. We also attempted to prolong the seed culture time, but the morphology had no significant change. The main concern to determine the seed culture time to be 28 h was to guarantee the seed in logarithmic growth phase, so shortening the seed culture cycle is not very desirable.

Point 12: Referring to 10^7 spores/mL as the control is questionable, as it is just a concentration as the others. Statistical comparison among the 4 initial spore concentrations should be calculated (e.g. with ANOVA with tukey post hoc) presented and presented in tables 1, 2, and 3 with superscripts that indicate significantly significant differences.

Response 12: Agreed, we changed the description. As to the spore concentrations, indeed, it’s better to provide a statistical comparison bewteen the four experiments. But the spore suspensions were diluted by gradient dilution, we missed to count the spore concentrations after dilution, thus it’s hard for us to provide the information about the exact concentrations of the spore suspensions.

Point 13: Tables 1, 2, 3: the legend should indicate the hours of the cultures. In table 2 it is not clear why the values are reported as function of S1, S2, S3, and S4 and not as function of 10^4, 10^5, 10^6, and 10^7 spores/mL.

Response 13: Thanks, we have added the culture time for the legends, and changed the information about colunm 1 in table 2.

Point 14: 187. What was the rational of selecting these values of spores/ml? It is not clear in what the optimization procedure consisted and the level of statistical significance of differences.

Response 14: In the previous section, we studied the yield difference under the condition of different orders of magnitude of inoculation quantities, and found that with the spore concentration of 10^5 spores/mL, the RML activity was relatively higher than others. To further optimize the process, the inoculated spore concentrations relatively lower and higher than 10^5 spores/mL (less than one order of magnitude) were chosen for further experiments.

Point 15: Figure 1: presumably values are means +/- SD. Please clarify and add indication of statistical significance

 Response 15: Agreed. The indication of statisticla significance has added into Figure 1.

Point 16: Figure 2 Legend: it is better to replace “rheological properties of fermentation broth” (that would implicate some quantitative result) with “mycelium morphology in cultures”.

Response 16: Thanks,we have changed it in the revised version.

Point 17: Section 3.2.2 could start at line 242

Response 17: Agreed.

Point 18: Table 4. It seems that the columns Spores concentrations should be entitled Pellets concentrations. The table and data description lacks statistical analysis, at least for comparing Mode 1 and Mode 2.

Response 18: Yes, it should be entitled Pellet concentrations. We have modified it in the revised version. The statistical analysis for comparing Mode 1 and Mode 2 was shown in Figue 4.

Point 19: Figures 3 and 4. Is there any information on the trend of the carbon source? It would be useful to better comprehend the status of the culture during the course of the fermentation run.

 Response 19: Agreed. The carbon source concentration was a valuable parameter for us to analyze the process. But the main carbon source used in this work was the corn dextrin,it’s hard to determine the exact concentration. What’s more, the corn dextrin can be efficiently ultilized by A. oryzae only when it was hydrolyze by enzymes to monosaccharides. We have tested the glucose concentration (which is a balance of producting and consumption) in the broth, but it was relatively low in the whole process and could hardly give valuable information for us.

Point 20: It should be better explained in the introduction the reason of reporting this section, that has not a lot to do the rest of the study that was carried out with a not-improved A. oryzae. The authors should clearly indicate was already known on the improved strains and why they included them in this manuscript that is mostly focused on the rheological behavior and in control of morphology. This section could be relevant if the mutants exhibited some morphology or behavior (e.g. formation of pellets) that could justify their increased performance.

Response 20: Agreed. It’s a great help for us to improve the logicality of the manuscript. Indeed, the problems involing adherent growth, aggregation and escape of liquid in the original strain also appeared in the high-yield strains. We have emphasized the problem in the manuscript.

Reviewer 2 Report

The authors of this manuscript describe the improvement of the productivity of heterologous lipase by Aspergillus oryzae. While research on improving the productivity of heterologous protein production by A. oryzae is critically important for industry, this manuscript contains following serious problems.

1. Although the authors claim to have achieved high production of Rhizomucor miehei lipase (RML) based on the values of enzyme activity, the method of this study cannot distinguish between the enzyme activity of RML and the A. oryzae original lipases. Therefore, not all values of enzyme activity presented by the authors can be considered to be due to RML.

2. The authors describe the relationship between RML production and morphological changes caused in different numbers of inoculated conidiospores. The relationship between morphological changes in A. oryzae and protein production level has been reported in previous studies such as Miyazawa et al Biosci Biotechnol Biochem (2016), Kurakake et al Fungal Biol (2020), and Ichikawa et al J Biosci Bioeng (2022), but this manuscript ignores them.

Minor comments

1. Scientific names of R. miehei, Pichia pastoris, and Trichoderme harzianum are not italicized.

2. Line 116, What is p-nitrophenylephene?

2. Line 117, Wang et al., 2012, shown as a reference, is missing from the reference list.

3. Line 117, Composition of substrate buffer is unknown.

4. Line 121, "as a control" means "as a blnak"?

5. Lines 142-143, No dilution factor indicated.

6. Line 175, Unclear what "control group" represents.

7. Table 4, Unclear what "Spore concentrations" represents.

Author Response

Point 1: Although the authors claim to have achieved high production of Rhizomucor miehei lipase (RML) based on the values of enzyme activity, the method of this study cannot distinguish between the enzyme activity of RML and the A. oryzae original lipases. Therefore, not all values of enzyme activity presented by the authors can be considered to be due to RML.

Response 1: Thanks for your suggestion. The original strain was provided by Wilmar Biotechnology R&D Center Co., Ltd, Shanghai, China. Previously, the strain was identified as Aspergillus oryzaes, and the lipase was identified as Rhizomucor miehei lipase. The expression of A. oryzae original lipase was extremely low in the Aspergillus oryzaes, similar to that reported in literatures. Furthermore, In order to improve the efficiency in assay of the enzyme activity, we chose a common method. We agreed that It would be better if we carried out a structure identification for the lipase after process optimization in 50 L bioreator. But, in our concetption, the process optimization can hardly change the type of lipase. What’s more, the morphology control strategies for process optimization in this work increased the lipase expression indeed.

Point 2: The authors describe the relationship between RML production and morphological changes caused in different numbers of inoculated conidiospores. The relationship between morphological changes in A. oryzae and protein production level has been reported in previous studies such as Miyazawa et al Biosci Biotechnol Biochem (2016), Kurakake et al Fungal Biol (2020), and Ichikawa et al J Biosci Bioeng (2022), but this manuscript ignores them.

Response 2: Thanks for your suggestion, it’s a great help to improve the manuscript. We have made more analysis about the literatures you mentioned in discussion part.

Point 3: Scientific names of R. miehei, Pichia pastoris, and Trichoderme harzianum are not italicized.

Response 3: Agreed. We have comprehensively checked the format and language issues in the whole text.

Point 4: Line 116, What is p-nitrophenylephene?

Response 4: Sorry, we made a mistake. It’s 4-Nitrophenyl palmitate in fact.

Point 5: Line 117, Wang et al., 2012, shown as a reference, is missing from the reference list.

Response 5: Thanks. The reference has been added to the reference list.

Point 6: Line 117, Composition of substrate buffer is unknown.

Response 6: The detailed composition of substrate buffer was added into the text.

Point 7: Line 121, "as a control" means "as a blnak"?

Response 7: Sorry, we made a confused description.

Point 8: Lines 142-143, No dilution factor indicated.

Response 8: The suspension was diluted with sterile water and the dilution factor was 1:9. The information has added into the text.

Point 9: Line 175, Unclear what "control group" represents.

Response 9: We have changed the confusing description in the revised version.

Point 10: Table 4, Unclear what "Spore concentrations" represents.

Response 10: Sorry, we made a mistake. It’s “Pellets concentrations” in fact.

Reviewer 3 Report

The manuscript looks interesting and reports potential new ways of regulating morphology to improve lipase production by A. oryzae. The authors reported they studied the effects of different spore concentrations on morphological changes and Rhizomucor miehei lipase (RML) production in flasks and 5 L bioreactors. They also evaluated different strains (previously produced by mutagenesis) that allowed to find one with a more efficient production of RML at the bioreactors. The entire study has some important data about lipase production at bioreactors. However it's important to highlight that is only the first step in developing novel bioreactors procedures. In addition, the authors described one highly efficient strain produced by mutagenesis, but they have not cited any previous study about them and have not even included a minimal description about the procedures to obtain the novel strains. So I think a full review is needed before resubmitting it.

As I described above, my main concern is that the manuscript reports that a high-yield strain was identified by mutagenesis that allowed to realize a highly efficient production of RML at the bioreactor level. However the authors did not include the methods and the results to obtain them and neither referred other previous studies. There is just one sentence: “high-yield strains (I7-6D-7, V7-6C-3, II4-4B-5, III3-5C-1) were obtained by ARTP mutagenesis". Even the abbreviation ARTP was not explained (“Atmospheric and room-temperature plasma”?). So I think it is necessary to describe the technique to obtain the "the high productivity strains" in the first section of the Methods as well as to report all strains obtained and selected by ARTP in the first Results section.  

On the other hand, it is not necessary to explain in detail the methods to Detect enzyme activity, protein content, sugar content and dry cell weight. All of them are well described in the scientific literature and the inclusion of the proper references is enough. Further, I would suggest the authors to include a flowchart with the sequential procedures of the Modes referred as 1 and 2. It would be very helpful to understand the whole study.

Despite the absence of the description of the selection of the strains, all other Results are quite well presented. But the Discussion is brief. I would suggest the authors to compare the performance of the selected “high productivity strains” with other previous studies. I also strongly suggest the author to include a paragraph with the limitations of the study, highlighting that it is necessary more studies to demonstrate the performance of these strains in industrial scale. The Conclusions section could also be more concise. It is not necessary to repeat the RML activities in all conditions (in U/mL). Please be more direct reporting the main general conclusions. 

Finally, the entire text also needs to be completely revised, including: the correction of the scientific names of the microorganisms genus and species that must be in italics. It is also necessary a description of all abbreviations used and a complete English grammar proofreading.

Author Response

Point 1: The manuscript looks interesting and reports potential new ways of regulating morphology to improve lipase production by A. oryzae. The authors reported they studied the effects of different spore concentrations on morphological changes and Rhizomucor miehei lipase (RML) production in flasks and 5 L bioreactors. They also evaluated different strains (previously produced by mutagenesis) that allowed to find one with a more efficient production of RML at the bioreactors. The entire study has some important data about lipase production at bioreactors. However it's important to highlight that is only the first step in developing novel bioreactors procedures. In addition, the authors described one highly efficient strain produced by mutagenesis, but they have not cited any previous study about them and have not even included a minimal description about the procedures to obtain the novel strains. So I think a full review is needed before resubmitting it.

Response 1: Thanks for your suggestions. We have totally revised the manuscript, including the addition of our previous work about the mutants.

Point 2: As I described above, my main concern is that the manuscript reports that a high-yield strain was identified by mutagenesis that allowed to realize a highly efficient production of RML at the bioreactor level. However the authors did not include the methods and the results to obtain them and neither referred other previous studies. There is just one sentence: “high-yield strains (I7-6D-7, V7-6C-3, II4-4B-5, III3-5C-1) were obtained by ARTP mutagenesis". Even the abbreviation ARTP was not explained (“Atmospheric and room-temperature plasma”?). So I think it is necessary to describe the technique to obtain the "the high productivity strains" in the first section of the Methods as well as to report all strains obtained and selected by ARTP in the first Results section.  

Response 2: Thanks for your suggestions. We have cited our previous work and explained the abbreviation ARTP in the revised version. The related information were added into the sections of the Methods and Results.

Point 3: On the other hand, it is not necessary to explain in detail the methods to Detect enzyme activity, protein content, sugar content and dry cell weight. All of them are well described in the scientific literature and the inclusion of the proper references is enough. Further, I would suggest the authors to include a flowchart with the sequential procedures of the Modes referred as 1 and 2. It would be very helpful to understand the whole study.

Response 3: Agreed. We have simplified the description of the common methods. For the methods which were sligtly changed, we made a detailed explanation. A flowchart with the sequential procedures of the Modes referred as 1 and 2 was shown in supplementary materials (Figure S1)

Point 4: Despite the absence of the description of the selection of the strains, all other Results are quite well presented. But the Discussion is brief. I would suggest the authors to compare the performance of the selected “high productivity strains” with other previous studies. I also strongly suggest the author to include a paragraph with the limitations of the study, highlighting that it is necessary more studies to demonstrate the performance of these strains in industrial scale. The Conclusions section could also be more concise. It is not necessary to repeat the RML activities in all conditions (in U/mL). Please be more direct reporting the main general conclusions. 

 Response 4: Thanks for your suggestions. Indeed, we need to point out the limitaions of this study and make a more direct conclusion. The detailed modifications were shown in the revised version.

Point 5: Finally, the entire text also needs to be completely revised, including: the correction of the scientific names of the microorganisms genus and species that must be in italics. It is also necessary a description of all abbreviations used and a complete English grammar proofreading.

 Response 5: Agreed. We have made a comprehensive proofreading and redressed the format and language issues. The detailed modifications were shown in the revised version.

Round 2

Reviewer 1 Report

Li et al. improved their manuscript and addressed most most issues of the previous version.

Perhaps the title could be simplified (e.g. Optimization of the fermentative production of Rhizomucor miehei lipase in Aspergillus oryzae by controlling morphology)

The style and English still has to be improved (prefer the past tense and impersonal sentences, avoid colloquial wording e.g. it's -> it was).

The statistic analysis is still scarce, especially for the values presented in tables. In the legends of tables and figures it should be indicated that values are means +/- SD (e.g. to interprete the error bars)

Lines 207 and 208 The sub-headings 3.2.1 3.2.2 seem not necessary and could be removed

Reviewer 2 Report

Point 1: Although the authors claim to have achieved high production of Rhizomucor

miehei lipase (RML) based on the values of enzyme activity, the method of this study cannot

distinguish between the enzyme activity of RML and the A. oryzae original lipases.

Therefore, not all values of enzyme activity presented by the authors can be considered to be

due to RML.

Response 1: Thanks for your suggestion. The original strain was provided by Wilmar

Biotechnology R&D Center Co., Ltd, Shanghai, China. Previously, the strain was identified as

Aspergillus oryzaes, and the lipase was identified as Rhizomucor miehei lipase. The expression of

A. oryzae original lipase was extremely low in the Aspergillus oryzaes, similar to that reported in

literatures. Furthermore, In order to improve the efficiency in assay of the enzyme activity, we

chose a common method. We agreed that It would be better if we carried out a structure

identification for the lipase after process optimization in 50 L bioreator. But, in our concetption,

the process optimization can hardly change the type of lipase. What’s more, the morphology

control strategies for process optimization in this work increased the lipase expression indeed.

I do not agree with the authors' claim that all the 4-nitrophenyl palmitate degrading activity is derived from RML. It cannot be ruled out that morphology control increases the production of A. oryzae original lipase(s) that degrades 4-nitrophenyl palmitate, and its activity may be included in the 4-nitrophenyl palmitate degradation activity that the authors have designated as RML activity. This is important point because the authors emphasize the high activity values of RML in this manuscript.

Reviewer 3 Report

As I mentioned in my first review, the manuscript looks interesting and reports potential new ways of regulating morphology to improve lipase production by A. oryzae. The authors reported they studied the effects of different spore concentrations on morphological changes and Rhizomucor miehei lipase (RML) production in flasks and 5 L bioreactors. In the previous version, they described different strains (previously produced by mutagenesis) that were evaluated. Now they refer to a previous study that selected one strain. So they only report the production of RML by that strain in the bioreactors. It's ok for me! The manuscript has also been relatively improved and presents more complete data and a better discussion of lipase production in bioreactors.

However, there are still issues in the manuscript that prevent this publication now. In Materials and Methods, I have already recommended including some appropriate references to the methodologies (“well described in the scientific literature”), but this has not been done. In the Results, it is even more serious, as the topic 3.2.1 (Morphology control strategy by optimizing the nitrogen source) is empty. What happened? In the Discussion, the authors did not include a paragraph with the limitations of the study. It is necessary!

Finally, the text needs to be fully proofread again for grammatical errors. Therefore, at least a minor revision is still required.

Round 3

Reviewer 2 Report

I understand the authors' response. However, we cannot know whether the lipase activity being measured in this study is specific for RML unless using a host strain not expressing RML as a control. I think the authors should replace the term ''RML activity'' with ''lipase activity''. In addition, I recommend that RML productivity be evaluated not only by lipase activity value, but also by the protein yield. For instance, RML yield can be simply examined by SDS-PAGE with some protein sample of known quantity and comparing the intensity of the bands.